# Neural Coding of Food Is a Multisensory, Sensorimotor Function

**DOI:** 10.3390/nu13020398

**Published:** 2021-01-27

**Authors:** Patricia M. Di Lorenzo

**Affiliations:** Department of Psychology, Binghamton University, Box 6000, Binghamton, NY 13902-6000, USA; dioren@binghamton.edu

**Keywords:** gustation, taste, food, neural coding, sensorimotor

## Abstract

This review is a curated discussion of the relationship between the gustatory system and the perception of food beginning at the earliest stage of neural processing. A brief description of the idea of taste qualities and mammalian anatomy of the taste system is presented first, followed by an overview of theories of taste coding. The case is made that food is encoded by the several senses that it stimulates beginning in the brainstem and extending throughout the entire gustatory neuraxis. In addition, the feedback from food-related movements is seamlessly melded with sensory input to create the representation of food objects in the brain.

## 1. Introduction

Sensory systems serve and reflect the ecological niche of the organism by actively curating those aspects of the environment promoting its survival and well-being. This idea is not new, and its application to the study of sensory systems has historically borne fruit, e.g., [1]. More recently, investigators in other sensory systems, including vision [2], somatosensation [3] and olfaction [4], have focused their attention on more naturalistic examples of sensation. In the study of the gustatory system, such an approach is relatively infrequent. Few experiments have focused on stimuli that emulate what an animal might encounter in its natural habitat, i.e., food. Instead, foods that are part of the subject’s diet have been distilled to a few “basic” taste qualities, namely sweet, salty, sour, bitter and umami. Decades of psychophysical data in both humans and animals have demonstrated the unambiguous singularity of these taste qualities, and dozens of studies have documented the neural responses to prototypical taste stimuli. The overarching experimental strategy has been the idea that, since the five basic taste qualities arguably span the sensory domain of taste, assessment of the neural representation of these stimuli would provide a nearly complete portrait of the landscape of the gustatory system. In contrast, our recent studies of the brainstem using evolutionarily relevant, and more complex, taste stimuli in awake, freely behaving animals have suggested that the response repertoire of the gustatory system is far richer than once believed. This repertoire includes response types that reflect the active process of acquiring information about food, including multimodal sensations (taste, olfaction, somatosensation) as well as ingestive behaviors. As a result, the coding strategies used by the brain to represent taste, and ultimately food, must be reimagined.

This review is a curated discussion of the relationship between the gustatory system and the perception of food beginning at the earliest stage of neural processing. It is not meant to be comprehensive. For example, the involvement of motivation and learning in the perceptual process surrounding taste/food are not discussed. A brief discussion of the idea of taste qualities and anatomy of the mammalian taste system is presented first, followed by an overview of theories of taste coding. Although our focus is on the gustatory system, we relate the multiple sensory modalities that are stimulated by food as well as the neural concomitants of motor behavior that accompany ingestion to the way that food is represented in the brain. We define a “taste stimulus” as a chemical that stimulates taste receptor proteins located on taste receptor cells in the taste buds. We define “flavor” as the combination of sensations, including taste, smell, texture, temperature, etc., that is evoked by a food. Finally, we explore a view of taste/food coding as a brainwide sensorimotor function.

## 2. Taste Qualities and Food—Defining the Landscape

A “taste quality” is defined as a group of chemical compounds that, first, evoke a taste sensation and, second, taste alike to humans and presumably to other mammals. The presumption that underlies the classification of a taste quality as “basic” is that they do not elicit any other taste quality and cannot be reproduced by any combination of other taste qualities [5]. There are five taste qualities that most researchers agree form the basis of the human taste system and that of most mammals, as mentioned above. Among other tests [6], the independence of these basic taste qualities has been evidenced by the absence of cross-generalization of conditioned taste aversions [7,8,9] in rodents. However, these studies did not rule out the possibility that other independent taste qualities exist. These include fat, calcium, starch and even water (reviewed in [6]).

The relationship of the basic taste qualities to food is a complex one. One might first consider how an animal, including a human, decides that any substance is edible. That is where other senses, in addition to taste, come into play. Vision, somatosensation and perhaps especially olfaction all serve critical roles in this function. Learning, no doubt, can modify what is eligible for consumption, adding to the adaptability of the organism to what is available, harmful or helpful as food. However, the metabolic state can modulate and overrule the judgement of what qualifies as a food. For example, what might be considered inedible to a sated animal would become eligible for consumption by a starving one [10]. Studies of “pica”, the voluntary ingestion of non-food substances, have also pointed to the metabolic state as a factor in the identification of edible substances [11,12,13]. However, as with the effects of learning, metabolic state may affect consumption, but the neural representation of a food in its multisensory complexity would presumably remain unaltered. That would preserve the ability to identify the substance being ingested. While it has been argued that the ultimate arbiter of what constitutes a food, as well as what determines intake, is its taste [14,15], the variety of sensations evoked by a food impact how it is identified, its palatability and ultimately the amount ingested. Thus, in considering how food is represented in the brain, the convergence of sensations must be in the mix. 

The collection of varied sensations evoked by food is collectively defined as its “flavor”. Conventional wisdom points to convergence of food-related sensations at high levels of the central nervous system, that is, in the cortex [16]. There are reports, however, both early and more recent, showing that brainstem nuclei that are traditionally thought to respond exclusively to taste stimuli also respond to touch [17,18], temperature [19,20] and odor [21,22]. Collectively, these results beg the question of how food-related signals from these non-gustatory senses interact with food-evoked taste signals at the earliest stages of central processing. Put another way, these observations point to the idea that the flavor of food, produced by the convergence of several modalities of sensation, may be represented at multiple levels of the central gustatory system.

## 3. Brief Description of the Anatomy of the Taste System in Mammals

Gustation in mammals begins with the taste receptor cells on the tongue. These cells are arranged like the sections of an orange in organs called “taste buds”. Taste buds are located in specialized papillae on the tongue’s surface. There are three types of papillae that contain taste buds: fungiform, which are round bumps located on the tip and sides of the rostral tongue; foliate, which are gill-like folds located on the sides of the tongue; and circumvallate, which are mushroom-shaped bumps located in an inverted V-shape at the back of the tongue. Rodents have only a single circumvallate papilla located at the back of the tongue. Sapid fluids interact with taste receptor proteins located on the apical villi of the taste receptor cells. The process of taste transduction has been recently reviewed in detail elsewhere [23]. 

Taste buds are innervated by three cranial nerves (CN) [24]. The chorda tympani (CT) branch of the facial nerve (CNVII) innervates taste buds on the rostral one-third of the tongue; the glossopharyngeal nerve (CNIX) innervates taste buds in the caudal one-third of the tongue and the vagus nerve (CNX) innervates the taste buds on the soft palate and epiglottis. Each of these nerves have their cell bodies in peripheral ganglia. The geniculate ganglia house cell bodies of the VIIth nerve. The IXth and Xth nerves are associated with the petrosal and nodose ganglia, respectively. In close association with the nerves that innervate the taste buds are fibers from the mandibular branch of the trigeminal nerve (CN5) [25,26]. This nerve conveys information about mechanical, thermal and irritating chemical sensations such as those evoked by capsaicin, the active ingredient in chili peppers. Cell bodies associated with the trigeminal nerve are located in the trigeminal ganglion. Interestingly, olfactory receptors in the taste buds have been described recently [27]. These data imply that the taste bud may convey more information about food than just taste. Thus, even at the level of the tongue, food is encoded by more than gustation alone.

Centrally, the three cranial nerves that innervate taste buds in the oropharyngeal area project in a roughly topographic pattern to the nucleus of the solitary tract (NTS). The projections of the VIIth nerve are at the most rostral tip with the projections of the IXth nerve, just caudal, with some overlap [24]. The most caudal projection field comes from the Xth nerve. Taste-responsive cells are located in the rostral central and medial portions of the NTS, while downstream projections to the reticular formation originate in the rostral ventral NTS [28].

In non-primate mammals, the NTS sends a major projection to the parabrachial nucleus of the pons (PbN). From the PbN, there are two pathways that carry information about taste upstream. The dorsal pathway travels to the parvicellular region of the ventroposteromedial thalamus (VPMpc) and to the primary gustatory cortex (GC) located in the agranular and dysgranular insula. The limbic pathway travels to the central nucleus of the amygdala, the lateral hypothalamus, the bed nucleus of the stria terminalis and the substantia innominata [29]. In primates, including humans, the central gustatory pathway bypasses the PbN and projects directly to the VPMpc and on to the GC [30]. As with all sensory systems, there is considerable centrifugal input to downstream structures [29], making the central gustatory system more like a collection of interacting loops rather than a feed forward linear pathway ending in the GC.

## 4. Neural Coding of Taste—The Raw Data

To place the various theories and perspectives of taste and/or food coding in context, it may be useful to consider what “raw material” the experimenter can use to construct models of brain processing and representation of a food. Figure 1A (spikes) and B (peristimulus–time histogram; PSTH) shows the firing pattern of two cells in the NTS in a urethane-anesthetized rat in response to a taste stimulus. For decades, researchers have measured such responses as the firing rate and/or number of spikes within some predetermined response interval. Several aspects of the response are immediately apparent: first, the response has a distinct time course with an initial phasic burst of firing followed by a more sustained, tonic elevation in firing rate. Simple response measures of spike count/rate ignore these response dynamics. Second, the response in these cells is long, lasting well past the taste stimulus presentation and in some cases well past the measured response interval used to determine response magnitude. Thus, response measures that rely on predetermined response intervals ignore the potential for variability in response length. Third, the response ends with the initiation of the water rinse. While most NTS cells in anesthetized animals show no response to water, some cells do [31], potentially confounding the measurement of response length. Figure 1C shows the response to a taste stimulus over ten trials of five licks each followed by water licks presented on a variable ratio 5 schedule (each water lick is followed and preceded by four to six unreinforced dry licks). Many of the same characteristics of the taste responses resemble those in anesthetized animals. However, the presentation of multiple stimulus trials illustrates the fact that taste responses can vary trial-by-trial.

In spite of these caveats, taste-evoked spike counts recorded from anesthetized animals have allowed comparisons of sensitivity across taste qualities and intensities in several species. In general, taste-responsive neurons respond to more than one of the basic taste qualities. Historically, using the same data, two different, but not mutually exclusive, theories of taste coding emerged and have dominated the literature. Both of these theories arose from studies of peripheral nerve responses to taste. These are the labeled line and across fiber or unit pattern theories. Proponents of both theories have been debating the merits and flaws of each approach for many decades.

The labeled line theory emphasizes the observation that taste-responsive neural elements fire most vigorously and consistently to a single taste quality when tastants are presented at midrange concentrations [32]. Most importantly, the identity of that “best stimulus” enables the reliable prediction of the “second best”, “third best” and so on. Thus, the idea emerged that there are fiber or neuron “types” associated with each of the basic taste qualities. The idea is that each group of fibers/neurons is solely responsible for encoding information about the best stimulus of that group. Responses to non-best, often called “sideband”, stimuli are considered noise, i.e., irrelevant. More recently, proponents of the labeled line theory have classified fibers/neurons as “specialists” or “generalists” based on the degree to which a fiber/neuron was narrowly tuned to a single taste quality [33]. Specialists are presumed to participate in taste quality labeled line coding, while generalists are thought to signal aversive tastes [33]. Early on, experimental manipulations such as conditioned taste aversion [34] and sodium deprivation [35,36] were shown to specifically affect the appropriate groups of best stimulus neuron types. More recently, genetic manipulations of receptor subtypes underlying sweet, bitter, salty and sour tastes have provided data supporting the labeled line theory in the peripheral [37] as well as the central nervous system [38]. In apparent contradiction of these results are data from Roper’s group [39] showing that taste cells in the geniculate ganglion (housing the cell bodies of the VIIth nerve) can change their best stimulus as well as their breadth of tuning, depending on the concentration of tastants tested.

In contrast to the labeled line theory, the across unit pattern theory eschews the idea of cell types to argue that taste stimuli are represented by the distributed arrangement of responsive cells in a given structure [40]. This theory takes advantage of the fact that the great majority of taste-responsive cells respond to more than one taste quality. Evidence in support of this theory is the high correlation of response magnitudes to similar tastants as well as low correlations of response magnitudes to dissimilar tastants. Interestingly, there is no need to propose a set of basic taste qualities with this theory. In fact, it has been argued that the taste world is characterized by a loose organization of taste stimuli, some of which are not easily categorized into groups representing the five basic taste qualities [40]. For example, Di Lorenzo et al. [41] showed that a conditioned taste aversion to ethanol did not generalize to any of the basic taste qualities, but instead generalized to a mixture of sweet and bitter tastants. Consistent with the across unit pattern theory, the existence of widely distributed broadly tuned neurons in the gustatory cortex has recently been described in alert mice using wide-field calcium imaging [42].

## 5. Taste as an Active, Sensorimotor Function

The application of taste coding theories based on data from the peripheral taste structures to data from central taste-related structures is problematical for two reasons. First, the labeled line and across unit pattern theories both ignore the response dynamics that occur in the central nervous system. Second, these theories limit their consideration to taste responses in taste-responsive cells. This may seem like an obvious point. However, there are a substantial number of cells located in so-called “taste relays” in the brain that do not respond to taste as well as cells that respond best to sensory modalities other than and perhaps in addition to taste, all of which may collaborate to convey information about taste/food.

To address taste response dynamics, a third coding strategy, called “temporal coding”, has been proposed. In one form, the temporal dynamics of firing in a taste response is thought to convey information about taste quality (reviewed in [43]). Results of several studies in anesthetized rats showed that when two tastants evoke similar firing rates, such that they cannot be discriminated from each other, temporal coding enables them to be disambiguated [44,45]. Evidence for a role for temporal coding in discrimination of the basic taste qualities [46], of different exemplars of the same taste qualities [47], different stimulus concentrations [48] and among taste mixtures [46] in brainstem neurons has been reported. Data from the NTS [49] and PbN [50] in awake, freely licking rats also show a contribution of spike timing to discrimination of taste quality. 

Another form of temporal coding, most often applied to cortical taste responses, consists of a predictable sequence of “states”, defined as an alignment of firing patterns across neurons, which signals taste quality [51]. The taste quality-specific sequence of states is stable across trials but the dwell time in each state varies trial to trial. This type of code can indicate taste quality [51] and can predict the occurrence of taste-evoked behaviors [52,53].

In addition to “traditional” taste-responsive cells, our lab has shown that there are multiple cell types in the brainstem of awake, freely licking rats that likely participate in taste coding. These include so-called “anti-lick” cells that reduce their firing rate while the animal is licking and “lick bout” cells that increase their firing rate when an animal initiates a lick bout. There are also numerous lick-related cells whose firing rate is “lick coherent,” i.e., it waxes and wanes along with licking, with peak firing rates occurring at different parts of the lick cycle. Despite the fact that lick-related cells do not overtly appear to respond to taste stimuli, their firing patterns nevertheless contribute information about taste quality [54]. Many taste-responsive cells also vary their firing rates according to the lick pattern, and some show lick coherence without any overt or obvious pairing to the lick pattern [50,54]. Moreover, both brainstem [54] and cortical [55] taste-related neural structures contain cells that track ingestive behavior, i.e., licking. Collectively, these data point to a fundamental connection between sensation and behavior in the taste system, making the characterization of taste coding sensorimotor rather than strictly sensory.

The concept that sensory coding incorporates both sensory and motor-related components underscores the idea that acquisition of sensory stimuli is an active process [56]. A critical link between behavior and the activation of sensory neurons has been shown in studies of somatosensation [57,58,59], vision [60] and olfaction [4,61]. In olfaction, for example, the dynamics of responses to odorants are shaped by the sniffing cycle in ways that can only be fully appreciated in awakened animals [61]. In gustation, the lick is thought to be the reference for encoding information about taste stimuli [62]. Moreover, in both olfaction and gustation, learning modifies both the motor and, consequently, sensory responses to a stimulus. For example, Gutierrez et al. [62] have shown that as learning a Go-no-Go task progresses, large networks of cells across structures are recruited to synchronize with the lick rhythm. Thus, the study of sensory coding in general and in the taste system in particular is best viewed in the context of motor behavior, that is, as an active, sensorimotor function. This view differs from previous conceptualizations of taste coding, including labeled line, across unit pattern and temporal coding theories in that it incorporates a contribution of the activity generated by active sensory acquisition as a key component of the code.

## 6. Multisensory Integration in the Gustatory System Is Fundamental to Encoding Information about Food

The intimate association between sensation and behavior and their simultaneous representation in areas traditionally thought to be solely sensory suggests that a laser focus on taste responsiveness in any taste-related brain region is missing half the picture. Both movements, or perhaps somatosensory feedback from the movements, and the sensations that result interact with each other to enhance and complete the perceptual experience. Movements associated with acquiring information about food would engender signals from a number of sensory modalities, not just taste. Thus, it is not surprising that all of the central nuclei in the taste system contain neurons that respond to sensory modalities other than taste, underscoring the well-known multisensory integration occurring in response to food. For example, there are neurons in the gustatory cortex that respond to olfactory [63,64,65,66], somatosensory, auditory and visual sensitivity [67] as well as gustatory stimuli. The gustatory portion of the thalamus contains neurons that respond to thermal [68], tactile [68,69], olfactory [70] and gustatory stimuli [68,69]. Likewise, in the gustatory brainstem, there are neurons that respond to thermal [19], tactile [71,72,73] and olfactory [21,22] stimuli. In every area, a portion of the taste cells also respond to one or more non-taste modalities. Other cells are specific to a single sensory modality.

Multimodal sensitivity across the gustatory neuraxis suggests that the gustatory system may be optimally tuned to naturally complex stimuli such as actual foods that engage a number of senses simultaneously rather than the prototypical exemplars of the basic taste qualities. We tested this idea by comparing the responses to traditional taste stimuli (sucrose for sweet, NaCl for salty, citric acid for sour and caffeine for bitter) to responses to their naturalistic counterparts (grape juice for sweet, clam juice for salty, lemon juice for sour and coffee for bitter). As we had hypothesized, we found that responses to naturalistic stimuli were more easily distinguishable from each other than responses to the standard array of tastants [74]. Another study showed that brainstem neurons convey more information about taste stimuli when they are paired with odors than when they are presented alone [21]. These studies support the idea that taste-responsive cells in general may respond best to stimuli that stimulate multiple food-related senses. However, they also show that this sort of multisensory convergence occurs at the very earliest stages of central processing of food, in the brainstem. Information about taste, texture and temperature is most likely derived directly from the oropharynx, but olfactory input almost certainly originates from top–down input. Figure 2 shows a summary of the structures and connections of the central taste and olfactory circuits. Although there are no known direct projections from the olfactory bulb or piriform cortex to the brainstem taste-related nuclei, olfactory input may be conveyed via the gustatory cortex [64], amygdala [75,76] or lateral hypothalamus [77]. This sort of convergence of information at an early stage enables rapid identification and ingestion/rejection decisions to be made.

## 7. Conclusions

For humans and animals, the perception of food begins before it enters the mouth. The sight and smell of food entice the eater toward ingestive movements. Once in the mouth, the taste, texture and retronasal smell of the food deliver sensations that fulfill the promise of the prescient signals. Feedback from food-related movements compliments and buttresses the other sensory inputs. In animal models, the most common approach to studying the neural representation of food has been to dissect its components and study each as an autonomous and independent contributor. However, a close examination of the response repertoires of the gustatory circuitry shows that the sensory components of food itself and the sensory feedback from movements associated with its consumption are processed simultaneously, both in parallel and as convergent input at all levels of the gustatory neuraxis, making the whole larger than the sum of its parts.

## Figures and Tables

**Figure 1 nutrients-13-00398-f001:**
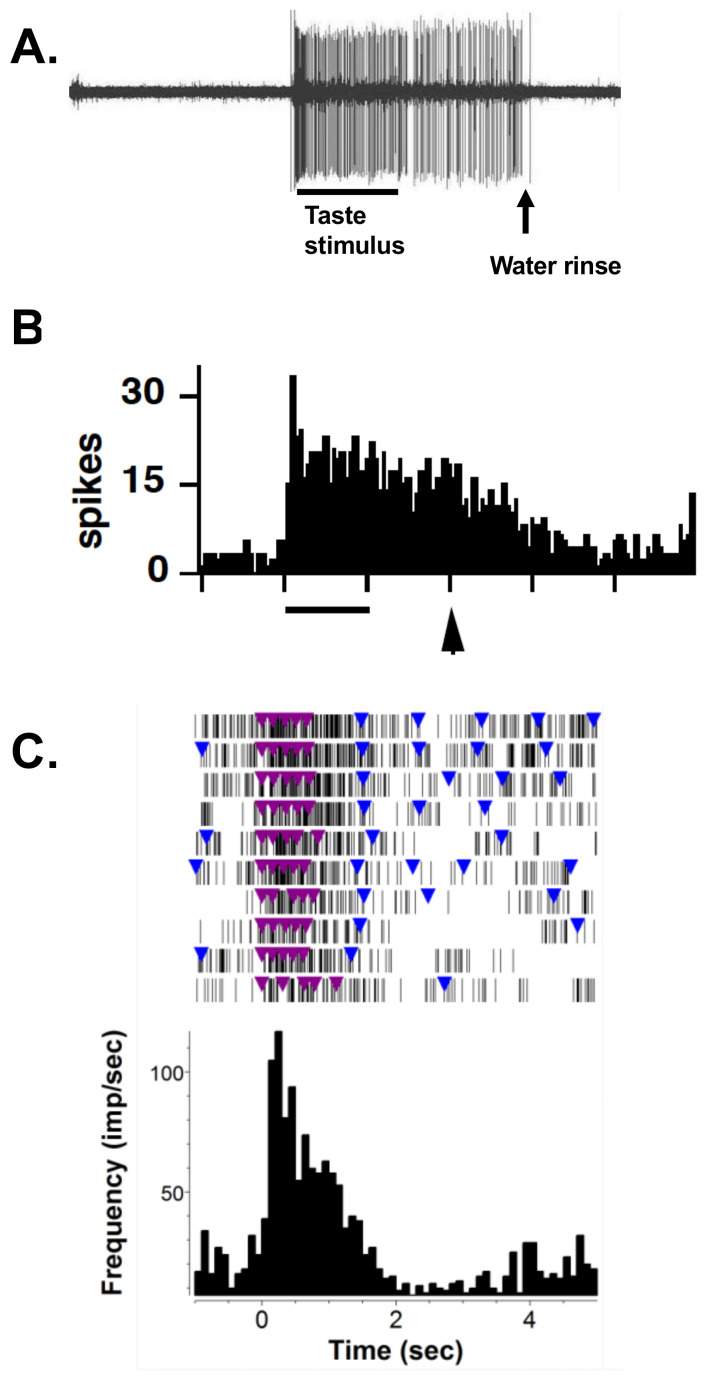
Examples of taste responses recorded from the nucleus of the solitary tract (NTS) in an anesthetized (**A**) and (**B**) and an awake, freely licking rat (**C**). (**A**) Firing pattern of a single NTS cell during the presentation of a taste stimulus for 5 s followed, after 5 s, by a water rinse in a urethane-anesthetized rat. (**B**) Peristimulus–time histogram (PSTH) showing the response during the presentation of a taste stimulus for 5 s (line under histogram) followed after 5 s by the presentation of a water rise for 20 s (arrow under histogram) in a urethane-anesthetized rat. (**C**) Raster (top) and PSTH (bottom) showing a taste response in an awake, freely licking rat. Colored triangles indicate licks: blue triangles show water licks; purple triangles show taste stimulus licks. Unreinforced licks between water licks are not shown. The taste stimulus trial consists of 5 consecutive stimulus licks. Ten trials are shown.

**Figure 2 nutrients-13-00398-f002:**
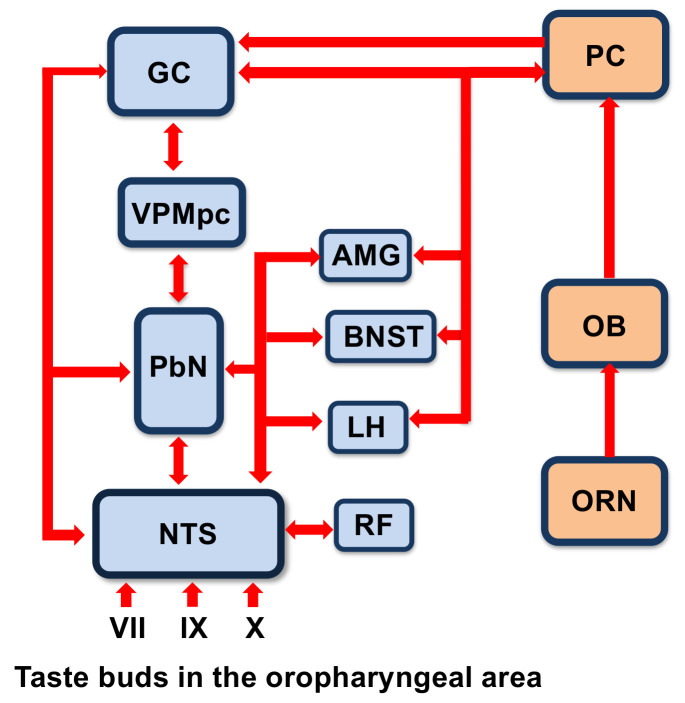
Diagram of the brain areas associated with gustation and olfaction showing their intersection. Abbreviations are as follows: VII, facial nerve; IX, glossopharyngeal nerve; X, vagus nerve; NTS, nucleus of the solitary tract; PbN, parabrachial nucleus of the pons; VPMpc, parvicellular region of the ventroposterolateral thalamus; GC, gustatory cortex; AMG, amygdala; BNST, bed nucleus of the stria terminalis; LH, lateral hypothalamus; RF, reticular formation; ORN, olfactory receptor neuron; OB, olfactory bulb; PC, piriform cortex. Arrows highlight known interconnections among structures.

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
