# Peer review of "Neural Coding of Food Is a Multisensory, Sensorimotor Function"

_nutrients, 2021, doi:10.3390/nu13020398_

Round 1
Reviewer 1 Report
This manuscript is a review about an interesting topic related to the gustatory system and taste processing. The content of the manuscript is valuable for researchers and professionals related to the anatomy and physiology of the gustatory system, as well as taste learning and the perception of flavors. I add here some comments that may improve the quality of the manuscript.
Title:
I am not sure if the title is clear enough about the content of the manuscript. With “brainwide” the author means brain network? Besides, if multimodal means multi-systems (motivational, emotional, memory, etc), is not sensorimotor one of these systems? I would recommend being more precise from the beginning, that is, from the title.
Abstract:
- From the abstract, the reader should know if the information provided about the gustatory system and taste processing refers to rodents, humans, mammals or vertebrates in general, or some other species.
Main text:
- Page 1: Introduction. After an introduction on the complexity of the gustatory system, I miss a mention about other systems involved in taste processing, as motivational, hedonic, emotional and memory systems. Taste processing is not only a sensorimotor process (please, consider this with respect to the title of the section 5). In addition, I still believe that a clarification about if the information refers to mammals or other species should be provided when necessary.
- Page 1 (line 43): The author states: “A “taste quality” is defined as a group of chemical compounds that taste alike”. I am not sure if this statement is enough precise. I would say that “A "taste quality" is that attributed to a group of chemical compounds evoking a taste”.
- Page 2 (lines 46-47): “There are five taste qualities that most researchers agree form the basis of our taste system…”. Again, what does mean “our” taste system? Does the author mean human or mammals system? More precision is desirable about this point.
A general problem in this section is the confusing mix of information about taste and food. It is not clear to me when the evidence reported refers to experiments on taste processing or it comes from food experiments. Definitely the information referring to a taste has to be differentiated from that referring to more complex taste stimuli (such as food).
- Page 2 (lines 47-48): “Among other tests [6], the independence of these basic taste qualities has been evidenced by the absence of cross-generalization of conditioned taste aversions”. This shows the importance of taste learning in taste processing. The author introduces this statement without any previous mention about the involvement of taste learning in taste processing. I would suggest an initial mention about the role of taste learning and memory before mixing this information with the description of basic tastes (or another kind of organization that does not neglect this point).
- Page 2 (lines 64-68): “While it has been argued that the ultimate arbiter of what constitutes a food, as well as what determines intake, is taste…” I would say flavor instead taste, as olfactory information is also determinant for eating decision.
- Page 2 (lines 69-70): “The collection of varied sensations evoked by food is collectively defined as its “flavor”. I would say evoked by taste stimuli, not necessarily by food. In rodents, for example, a single taste stimulus is enough to evoke a flavor.
- Page 2 (lines76-77): “the flavor of food may represented at multiple levels of the central gustatory system”. Something is wrong here. Does the author mean “the flavor of food may be represented at multiple levels of the central gustatory system”. By the way, I would say “the flavor of taste stimuli may be represented…”.
- Page 2 (line 90, section 3): “…the glossopharyngeal nerve innervates taste buds…” Since the author specifies the number of the facial (VII) and vagus (X) cranial nerves, why not is doing the same with the glossopharyngeal (IX) nerve?
- Page 3 (lines113-114): “The limbic pathway travels to the central nucleus of the amygdala…”. And to the basolateral amygdala too (Molero-Chamizo A, Rivera-Urbina GN. Taste Processing: Insights from Animal Models. Molecules. 2020 Jul 8;25(14):3112. doi: 10.3390/molecules25143112. PMID: 32650432; PMCID: PMC7397205). Again, these data come mainly from rodent model.
- Page 3 (lines 123-124): “Figure. 1A (spikes) and B (peri-123 stimulus-time histogram; PSTH) shows the firing pattern of two cells in the NTS in a ure-124 thane-anesthetized rat in response to a taste stimulus”. Please, provide here the reference/s supporting these data (also in the legend of Figure 1). In the legend of Figure 1: “…water licks rare not shown”. Does the author mean “are not shown”? By the way, I would say “during the presentation of a taste stimulus for 5 s” instead of “during a 5 s taste stimulus” (same throughout the entire legend).
- Page 3 (lines 137-138): “…on a variable ratio 5 schedule”. Does the author mean 5 s schedule?
- Page 5 (section 4, lines 188-189): It is no clear to me what the author means with “…taste is more like olfaction in that there is only a loose organization of like stimuli with many stimuli difficult to characterize”. It might be necessary a better explanation to understand what the message is here.
- Page 5 (section 5, line 213): “Data from the NTS [49] and PbN [50] awake, freely licking rats…”. Something is missing here… Does the author mean “Data from the NTS [49] and PbN [50] in awake, freely licking rats…”?
- Page 6 (section 6, line 265): In the last part of the first paragraph all information is provided in present, so what is the sense for using the past tense in the last sentences?:” In every area, a portion of the taste cells also responded to one or more non-taste modalities. Other cells were specific to a single sensory modality”
- Figure 2: LH abbreviation is not mentioned in the legend. On the other hand, why with the cranial nerves it only appears the word “tongue”? The vagus nerve innervates the pharynx…
- Conclusions: it seems to me that “which” does not make sense in this statement: “The most common approach to studying the neural representation of food which has been to dissect its components and study each as an autonomous and independent contributor”. Please, review the sense of that.
- Conclusions: again, the reader does not know in this point if the author is referring to humans (“deliver sensations that fulfill the promise of the prescient signals”), rodents (“dissect its components and study each as an autonomous and independent contributor”), or a mix of both. Please, clarify when information come from humans or other species. Otherwise, the reading is confusing.
- General comment: The author may consider the possibility to add a brief resume o general idea at the end of each section. This would avoid the feeling of the accumulation of information through the reading without reaching principal ideas.
Author Response
Reviewer #1
This manuscript is a review about an interesting topic related to the gustatory system and taste processing. The content of the manuscript is valuable for researchers and professionals related to the anatomy and physiology of the gustatory system, as well as taste learning and the perception of flavors. I add here some comments that may improve the quality of the manuscript.
Title:
I am not sure if the title is clear enough about the content of the manuscript. With “brainwide” the author means brain network? Besides, if multimodal means multi-systems (motivational, emotional, memory, etc), is not sensorimotor one of these systems? I would recommend being more precise from the beginning, that is, from the title.
RESPONSE: I changed the title to “Neural coding of food is a multisensory, sensorimotor function”
Abstract:
- From the abstract, the reader should know if the information provided about the gustatory system and taste processing refers to rodents, humans, mammals or vertebrates in general, or some other species.
RESPONSE: The word “mammalian” was added to Line 9 of the Abstract to clarify that the discussion refers to mammals, as opposed to invertebrates, birds, etc.
Main text:
- Page 1: Introduction. After an introduction on the complexity of the gustatory system, I miss a mention about other systems involved in taste processing, as motivational, hedonic, emotional and memory systems. Taste processing is not only a sensorimotor process (please, consider this with respect to the title of the section 5). In addition, I still believe that a clarification about if the information refers to mammals or other species should be provided when necessary.
RESPONSE: I added the following sentence on Lines 39-40: “So, for example, the involvement of motivation and learning in the perceptual process surrounding taste/food will not be discussed.” Since this is a rather restricted review, mostly centered around the mammalian brainstem, a full discussion of these other systems, though obviously important, is beyond the scope of the manuscript. In addition, I would argue that these higher-order systems do not affect basic perceptual attributes of a taste stimulus or food. For example, although an animal might learn to avoid a sweet stimulus, it can still identify it as sweet. A similar argument can be made about the motivation to consume a given food…identification is primary, followed secondarily by a decision to consume. I also clarified that the discussion of the taste system as well as taste coding refers to “mammals” on Line 41.
- Page 1 (line 43): The author states: “A “taste quality” is defined as a group of chemical compounds that taste alike”. I am not sure if this statement is enough precise. I would say that “A "taste quality" is that attributed to a group of chemical compounds evoking a taste”.
RESPONSE: I have modified the opening sentence to this sections as follows: “A “taste quality” is defined as a group of chemical compounds that first, evoke a taste sensation and second, taste alike to humans and presumably to other mammals.” Lines 45-46.
- Page 2 (lines 46-47): “There are five taste qualities that most researchers agree form the basis of our taste system…”. Again, what does mean “our” taste system? Does the author mean human or mammals system? More precision is desirable about this point.
RESPONSE: I clarified which species identify the five basic taste qualities as “humans and most mammals” on Line 50. I say “most” mammals because, for example, cats do not taste chemicals that humans label as sweet. I did not feel that including this qualification was relevant to this discussion so I did not include it.
A general problem in this section is the confusing mix of information about taste and food. It is not clear to me when the evidence reported refers to experiments on taste processing or it comes from food experiments. Definitely the information referring to a taste has to be differentiated from that referring to more complex taste stimuli (such as food).
RESPONSE: To address this issue, I have clarified what I mean by taste processing versus food processing in the two concluding sentences of this section. They read as follows: “Collectively, these results beg the question of how food-related signals from these non-gustatory senses interact with food-evoked taste signals at the earliest stages of central processing. Put another way, these observations point to the idea that the flavor of food, produced by the convergence of several modalities of sensation, may be represented at multiple levels of the central gustatory system.”
- Page 2 (lines 47-48): “Among other tests [6], the independence of these basic taste qualities has been evidenced by the absence of cross-generalization of conditioned taste aversions”. This shows the importance of taste learning in taste processing. The author introduces this statement without any previous mention about the involvement of taste learning in taste processing. I would suggest an initial mention about the role of taste learning and memory before mixing this information with the description of basic tastes (or another kind of organization that does not neglect this point).
RESPONSE: Please see response to Comment 1. above.
- Page 2 (lines 64-68): “While it has been argued that the ultimate arbiter of what constitutes a food, as well as what determines intake, is taste…” I would say flavor instead taste, as olfactory information is also determinant for eating decision.
RESPONSE: The second part of this sentence acknowledges the influence of olfaction as well as other senses: the variety of sensations evoked by a food impact how it is identified, its palatability and ultimately the amount ingested.” So, I am not sure that changing the word “taste” to “flavor” would be necessary or meaningful in this context.
- Page 2 (lines 69-70): “The collection of varied sensations evoked by food is collectively defined as its “flavor”. I would say evoked by taste stimuli, not necessarily by food. In rodents, for example, a single taste stimulus is enough to evoke a flavor.
RESPONSE: Respectfully, a “taste stimulus” by definition just stimulates taste receptors. It is only food that has flavor because of the variety of sensory modalities that it stimulates.
- Page 2 (lines76-77): “the flavor of food may represented at multiple levels of the central gustatory system”. Something is wrong here. Does the author mean “the flavor of food may be represented at multiple levels of the central gustatory system”. By the way, I would say “the flavor of taste stimuli may be represented…”.
RESPONSE: “may represented” has been changed to “may be represented” Thanks for catching this. Again, taste stimuli per se do not evoke flavors. By definition taste stimuli stimulate the gustatory system. Other attributes of a taste stimulus, its odor or texture for example, may evoke a flavor. However, the tastant itself is transduced and encoded as a solely gustatory sensation.
- Page 2 (line 90, section 3): “…the glossopharyngeal nerve innervates taste buds…” Since the author specifies the number of the facial (VII) and vagus (X) cranial nerves, why not is doing the same with the glossopharyngeal (IX) nerve?
RESPONSE: I have added the number of cranial nerve nine to the text as “glossopharyngeal nerve (CNIX)” as requested. I also changed the numbers of the seventh and tenth cranial nerves to Roman numerals to be consistent with the remainder of the paragraph.
- Page 3 (lines113-114): “The limbic pathway travels to the central nucleus of the amygdala…”. And to the basolateral amygdala too (Molero-Chamizo A, Rivera-Urbina GN. Taste Processing: Insights from Animal Models. Molecules. 2020 Jul 8;25(14):3112. doi: 10.3390/molecules25143112. PMID: 32650432; PMCID: PMC7397205). Again, these data come mainly from rodent model.
RESPONSE: As far as I know, the taste-responsive portion of the PbN only projects to the central nucleus, though there are extensive reciprocal connections between the central and basolateral nuclei. The reference that you cite talks about the connection of the gustatory cortex and the basolateral nucleus. See a recent review: Vincis R, Fontanini A. Central taste anatomy and physiology. Handb Clin Neurol. 2019;164:187-204. doi: 10.1016/B978-0-444-63855-7.00012-5. PMID: 31604547; PMCID: PMC6989094.
- Page 3 (lines 123-124): “Figure. 1A (spikes) and B (peri-123 stimulus-time histogram; PSTH) shows the firing pattern of two cells in the NTS in a ure-124 thane-anesthetized rat in response to a taste stimulus”. Please, provide here the reference/s supporting these data (also in the legend of Figure 1). In the legend of Figure 1: “…water licks rare not shown”. Does the author mean “are not shown”? By the way, I would say “during the presentation of a taste stimulus for 5 s” instead of “during a 5 s taste stimulus” (same throughout the entire legend).
RESPONSE: For Figure 1 A and B, the responses that are shown have never been published. I replaced Figure 1B because the old Figure 1B was from an old paper of mine for 2006 and I hadn’t realized it. So, I found another response that had never been published and included it as Figure 1B. I also corrected the typo from “rare” to ”are” in the figure legend. I also changed the Figure legend to reword the descriptions of the stimulus presentations as follows:
Figure 1. examples of taste responses recorded from the NTS in an anesthetized (A. and B.) and awake, freely licking rat (C.). A. Firing pattern of a single NTS cell during the presentation of a taste stimulus for 5 s followed, after 5 s, by a water rinse in a urethane-anesthetized rat. B. PSTH showing the response during the presentation of a taste stimulus for 5 s (line under histogram) followed after 5 s by the presentation of a water rise for 20 s (arrow under histogram) in a urethane-anesthetized rat. C. Raster (top) and PSTH (bottom) showing a taste response in an awake, freely-licking rat. Colored triangles indicate licks: blue triangles show water licks; purple triangles show taste stimulus licks. Unreinforced licks between water licks are not shown. The taste stimulus trial consists of 5 consecutive stimulus licks. Ten trials are shown.
- Page 3 (lines 137-138): “…on a variable ratio 5 schedule”. Does the author mean 5 s schedule?
RESPONSE: I did mean a VR5 schedule. I clarified this in the text as follows: “variable ratio 5 schedule (each water lick is followed and preceded by four to six unreinforced dry licks).”
- Page 5 (section 4, lines 188-189): It is no clear to me what the author means with “…taste is more like olfaction in that there is only a loose organization of like stimuli with many stimuli difficult to characterize”. It might be necessary a better explanation to understand what the message is here.
RESPONSE: In response to the reviewer’s concerns, I modified the text by deleting the reference to olfaction and clarifying the concept that taste stimuli are not easily categorized into groups. THE specific text is a follows: “In fact, it has been argued that the taste world is characterized by a loose organization of taste stimuli, some of which are not easily categorized into groups representing the five basic taste qualities [40].”
- Page 5 (section 5, line 213): “Data from the NTS [49] and PbN [50] awake, freely licking rats…”. Something is missing here… Does the author mean “Data from the NTS [49] and PbN [50] in awake, freely licking rats…”?
RESPONSE: I added the word “in” as suggested. Thanks.
- Page 6 (section 6, line 265): In the last part of the first paragraph all information is provided in present, so what is the sense for using the past tense in the last sentences?:” In every area, a portion of the taste cells also responded to one or more non-taste modalities. Other cells were specific to a single sensory modality”
RESPONSE: Good point. I have changed the last sentence of that paragraph to to read: “In every area, a portion of the taste cells also respond to one or more non-taste modalities. Other cells are specific to a single sensory modality.”
- Figure 2: LH abbreviation is not mentioned in the legend. On the other hand, why with the cranial nerves it only appears the word “tongue”? The vagus nerve innervates the pharynx…
RESPONSE: Good catch. Thanks! I have added: “LH, lateral hypothalamus” to the legend of Figure 2. I have also changed Figure 2, replacing “tongue” with “taste buds in the oropharyngeal area.”
- Conclusions: it seems to me that “which” does not make sense in this statement: “The most common approach to studying the neural representation of food which has been to dissect its components and study each as an autonomous and independent contributor”. Please, review the sense of that.
RESPONSE: Another good catch. Thanks! This was a typo. I have removed the word “which” from that sentence.
- Conclusions: again, the reader does not know in this point if the author is referring to humans (“deliver sensations that fulfill the promise of the prescient signals”), rodents (“dissect its components and study each as an autonomous and independent contributor”), or a mix of both. Please, clarify when information come from humans or other species. Otherwise, the reading is confusing.
RESPONSE: I have added a clarification as follows: For humans and animals, the perception of food begins before it enters the mouth. The sight and smell of food entice the eater toward ingestive movements. Once in the mouth, the taste, texture and retronasal smell of the food deliver sensations that fulfill the promise of the prescient signals. Feedback from food-related movements compliment and buttress the other sensory inputs. In animal models, the most common approach to studying the neural representation of food has been to dissect its components and study each as an autonomous and independent contributor. However, a close examination of the response repertoires of the gustatory circuitry shows that the sensory components of food itself and the sensory feedback from movements associated with its consumption are processed simultaneously, both in parallel and as convergent input at all levels of the gustatory neuraxis, making the whole larger than the sum of its parts.”
- General comment: The author may consider the possibility to add a brief resume o general idea at the end of each section. This would avoid the feeling of the accumulation of information through the reading without reaching principal ideas.
RESPONSE: Respectfully, I am confused by this comment/suggestion. When I tried to comply, I found that the title of each section was a summary of the main idea. Does the Reviewer wish me to reiterate that at the end of every section? The Conclusions section sums up the main points of the entire manuscript. I am really unsure how to respond.
Reviewer 2 Report
This review article provides a curated discussion of the relationship between the gustatory system and the perception of food. The author proposes some interesting ideas, such that the neural taste system is best viewed as an active, sensorimotor function. Such new ideas may be worth reporting.
However, there are some concerns with the current manuscript.
- It is unclear whether the concept of taste as an active, sensorimotor function could be the newly proposed concept or the minor modification of the previous temporal coding theory (l. 205). I recommend that the author bring the contrast between previous and new concepts.
- The discussion of multimodal integration (l. 248) is ambiguous. First, is this idea new? As the author reviews (l. 71), several previous studies have reported that the brainstem nuclei’s multimodal sensitivity. Second, it is unclear whether motor components (l. 235) or sensory signals produced by movements (l. 255) could be relevant to an active, sensorimotor function. Third, the discussion of top-down input (l. 281) lacks evidence and specificity. Finally, the information is Figure 2 is unclear; is it the summary of already know anatomical connectivity or the speculated functional circuit? I recommend that the author clarify these issues on the discussion of multimodal integration or focus on the discussion of an active, sensorimotor function.
Author Response
Reviewer #2
This review article provides a curated discussion of the relationship between the gustatory system and the perception of food. The author proposes some interesting ideas, such that the neural taste system is best viewed as an active, sensorimotor function. Such new ideas may be worth reporting.
However, there are some concerns with the current manuscript.
- It is unclear whether the concept of taste as an active, sensorimotor function could be the newly proposed concept or the minor modification of the previous temporal coding theory (l. 205). I recommend that the author bring the contrast between previous and new concepts.
RESPONSE: Lines 285-288. I added an explanation to address this concern. Specifically: “This view differs from previous conceptualizations of taste coding, including labeled line, across unit pattern and temporal coding theories, in that it incorporates a contribution of the activity generated by active sensory acquisition as a key component of the code.”
- The discussion of multimodal integration (l. 248) is ambiguous. First, is this idea new? As the author reviews (l. 71), several previous studies have reported that the brainstem nuclei’s multimodal sensitivity.
RESPONSE: As stated in the text, multimodal integration has been documented in every nucleus in the main gustatory pathway. We refer to the “well-known multisensory integration” in the first paragraph of that section. So, technically, the idea of multisensory integration is not new, but the incorporation of movement-related activity into the neural code is the mini contribution, as stated earlier (see prior comment).
Second, it is unclear whether motor components (l. 235) or sensory signals produced by movements (l. 255) could be relevant to an active, sensorimotor function.
RESPONSE: In our response to concerns about differentiating the ides put forth in the manuscript from the existing taste coding theories, I added a statement clarifying how the association of sensory and motor-related activity were key components to the neural code for taste/food. See statement above. In addition, I stated on Line 293: “Both movements, or perhaps somatosensory feedback from the movements, and the sensations that result interact with each other to enhance and complete the perceptual experience.”
Third, the discussion of top-down input (l. 281) lacks evidence and specificity.
RESPONSE: The statement suggesting that olfactory input to the brainstem taste-related nuclei is derived from top-down input is based on the lack of known direct projections from either the olfactory bulb or piriform cortex. I have added an explanation as follows: “Although there are no known direct projections from the olfactory bulb or piriform cortex to the brainstem taste-related nuclei, olfactory input may be conveyed via the gustatory cortex [65], amygdala [77,78] or lateral hypothalamus [79].”
Finally, the information is Figure 2 is unclear; is it the summary of already know anatomical connectivity or the speculated functional circuit? I recommend that the author clarify these issues on the discussion of multimodal integration or focus on the discussion of an active, sensorimotor function.
RESPONSE: These connections are all well known. To clarify this, I included a sentence as follows: “Figure 2 shows a summary of the structures and connections of the central taste and olfactory circuits.”